# Probing the Role of the Hinge Segment of Cytochrome P450 Oxidoreductase in the Interaction with Cytochrome P450

**DOI:** 10.3390/ijms19123914

**Published:** 2018-12-06

**Authors:** Diana Campelo, Francisco Esteves, Bernardo Brito Palma, Bruno Costa Gomes, José Rueff, Thomas Lautier, Philippe Urban, Gilles Truan, Michel Kranendonk

**Affiliations:** 1Center for Toxicogenomics and Human Health, Genetics, Oncology and Human Toxicology, NOVA Medical School, Faculdade de Ciências Médicas, Universidade Nova de Lisboa, 1150-082 Lisbon, Portugal; diana.campelo@nms.unl.pt (D.C.); francisco.esteves@nms.unl.pt (F.E.); bernardo.palma@nms.unl.pt (B.B.P.); bruno.gomes@nms.unl.pt (B.C.G.); jose.rueff@nms.unl.pt (J.R.); 2LISBP, Université de Toulouse, CNRS, INRA, INSA, 31077 Toulouse CEDEX 04, France; lautier@insa-toulouse.fr (T.L.); philippe.urban@insa-toulouse.fr (P.U.); gilles.truan@insa-toulouse.fr (G.T.)

**Keywords:** NADPH-cytochrome P450 reductase (CPR), microsomal cytochrome P450 (CYP), Cytochrome *b*_5_ (CYB5), protein dynamics, electron-transfer (ET), protein–protein interaction

## Abstract

NADPH-cytochrome P450 reductase (CPR) is the unique redox partner of microsomal cytochrome P450s (CYPs). CPR exists in a conformational equilibrium between open and closed conformations throughout its electron transfer (ET) function. Previously, we have shown that electrostatic and flexibility properties of the hinge segment of CPR are critical for ET. Three mutants of human CPR were studied (S243P, I245P and R246A) and combined with representative human drug-metabolizing CYPs (isoforms 1A2, 2A6 and 3A4). To probe the effect of these hinge mutations different experimental approaches were employed: CYP bioactivation capacity of pre-carcinogens, enzyme kinetic analysis, and effect of the ionic strength and cytochrome *b*_5_ (CYB5) on CYP activity. The hinge mutations influenced the bioactivation of pre-carcinogens, which seemed CYP isoform and substrate dependent. The deviations of Michaelis-Menten kinetic parameters uncovered tend to confirm this discrepancy, which was confirmed by CYP and hinge mutant specific salt/activity profiles. CPR/CYB5 competition experiments indicated a less important role of affinity in CPR/CYP interaction. Overall, our data suggest that the highly flexible hinge of CPR is responsible for the existence of a conformational aggregate of different open CPR conformers enabling ET-interaction with structural varied redox partners.

## 1. Introduction

Microsomal cytochrome P450 (CYP) metabolism requires a coupled supply of electrons, which are donated by the auxiliary protein NADPH cytochrome P450 oxidoreductase (CPR). CPR, encoded by the *POR* gene, is a ~78-kDa electron-transferring diflavin enzyme anchored to the membrane of the endoplasmic reticulum [1]. CPR mediates a two-electron transfer (ET) per reaction cycle, originated from NADPH enabling CYP-mediated metabolism of many compounds. These include endobiotics, e.g., steroids, bile acids, vitamins and arachidonic acid metabolites, as well as many xenobiotics, including therapeutic drugs and environmental toxins [2,3]. Moreover, CPR is the unique electron supplier of heme oxygenase, squalene monooxygenase and fatty acid elongase [4], sustaining exclusively the activity of these enzymes. Cytochrome *b*_5_ (CYB5) can donate the second electron to CYP, competing with CPR for the binding site on the proximal side of CYP [5]. CYB5’s interaction may have either a stimulating, inhibiting or having no effect over CYP catalytic activity, which seems to be CYP isoform and even substrate dependent [6].

CPR comprises a number of structurally distinct domains namely an N-terminal hydrophobic membrane anchoring domain; two flavin binding domains for flavin adenine dinucleotide (FAD) and flavin mononucleotide (FMN); a linker domain joining the FMN and FAD domains, as the provider of structural flexibility; and an NADPH binding domain [7]. The hinge segment, a highly flexible stretch with no defined secondary structure links the FMN and the connecting/FAD domain [8,9,10]. Electrons are transferred from NADPH through FAD (reductase) and FMN (transporter) coenzymes of CPR to redox partners, such as to the heme group in the reactive center of CYP [11].

Initial structural studies of CPR identified compact conformations that allowed internal (inter-flavin) ET, but were unable to reduce external acceptors [8,12,13]. Subsequently, three separate studies identified different open structures of CPR that allowed ET to redox partners, indicative of domain motion of CPR [10,14,15]. It is now fairly established that CPR exists in a conformational equilibrium between open and closed states in its ET function, which is highly dependent on ionic strength conditions [10,16,17]. The transition between these states appears to occur through a rapid swinging and rotational movement [17,18]. Certain residues in the hinge region have been suggested to be of importance for these large conformational changes [19], and seem to form a conformational axis, involved in a partial rotational movement of the FMN domain relative to the remainder of the protein [10,18].

Analysis of CPR domain dynamics is pertinent to understand its role in the interactions with its natural redox partners and its gated ET function. The affinities between CPR and CYP have been indicated among the factors modulating the protein dynamics of CPR. Different CYP isoforms may be differently served by CPR gating its ET differentially [20,21]. Although advances obtained during the last decade, CPR’s structural features controlling ET are not yet properly identified. CPR mutations may perturb specific structural requisites, necessary for the optimal transition between open and closed conformations, as well as disturb the interaction of CPR with its redox partners [10,20,21,22].

Previously, we have studied the effect of mutations in CPR on its redox partners [20,22,23] and the effect of alterations in the hinge segment in CPR-dependent cytochrome *c* reduction [24]. These hinge mutations showed differential effects on the conformational equilibrium of CPR and ET efficiency to cytochrome *c*, a non-physiological redox partner of CPR. Through modulation of the ionic strength conditions we demonstrated that electrostatic and flexibility properties of the hinge are critical for ET function, in which CPR’s membrane anchoring was shown to play an important role [24]. Although frequently used as a surrogate, the soluble cytochrome *c* has been indicated to interact differently with CPR, when compared with interactions of natural membrane-bound partners, such as CYP [25]. The use of cytochrome *c* as redox partner may have obscured additional important clues on structural features of the hinge segment involved in CPR’s open/closed dynamics and its gated ET function. To address this issue, three hinge mutants were selected from the initial eight mutants of our former study, based on their specific phenotypes in cytochrome *c* reduction. Human membrane bound CPR mutants S243P, I245P and R246A (numbering according to the human CPR consensus amino acid sequence NP_000932) were each combined with three different human CYPs, namely CYP1A2, 2A6 or 3A4, representatives of three major CYP families involved in drug metabolism [2,3]. The effect of the structural deviations of the three mutants was probed to obtain further insights on the role of the hinge segment of CPR in the interaction and ET with these physiological redox partners, using different experimental approaches.

## 2. Results

### 2.1. Bacterial Coexpression of CPR Mutants and CYP

Wild-type CPR and CPR hinge mutants S243P, I245P and R246A were separately introduced in the *E. coli* BTC strain and co-expressed with CYP1A2, 2A6 or 3A4, using methods described previously [26,27]. CYP expression levels were determined in bacterial whole-cells (Table 1). When co-expressed with CPR variants, CYP expression levels varied between 109–241 nM, 96–130 nM and 105–143 nM for CYP1A2, 2A6 and 3A4, respectively. Expression levels for these CYPs were comparable with those found previously with BTC strains [22,28,29]. More importantly, no large deviations were found in the CPR:CYP ratio between the four CPR variants, when expressed with each of the three CYPs (see Table 1). This enabled us to ascribe differences in activities of the CPR variants to the mutations, and not to variations in the stoichiometry between the two enzymes. These ratios were actually similar to those observed in our previous studies [22,28,29] and are in the range of those observed in human liver microsomes [30,31].

### 2.2. CYP-Activities When Combined with the Three CPR Hinge Domain Mutant

#### 2.2.1. Whole-Cell Bioactivation Assays

A whole cell/bioactivation assay was used for the first evaluation of the effect of the three hinge mutations on the activity of the three CYPs. This approach made use of the applicability of the BTC-CYP bacteria in mutagenicity testing [26,28]. The levels in CYP-dependent bioactivation of different pre-carcinogens, namely 2AA (2-aminoanthracene), IQ (2-amino-3-methylimidazo(4,5-*f*)quinolone), NNdEA (*N*-nitrosodiethylamine), NNK (4-(methylnitrosamino)-1-(3-pyridyl)-1-butanone) and AfB1 (aflatoxin B1)) were determined (Table 2; Figure 1). Interestingly, two of the three CPR hinge mutants lead to bioactivation capacities, which were either stimulated or equal, in comparison when CYPs were sustained by WT CPR, except for mutant I245P. This hinge mutant demonstrated a decrease for CYP1A2 and CYP3A4 mediated bioactivation of 2AA and AfB1, respectively. In contrast, CYP1A2 and CYP2A6 bioactivation capacities (for IQ and NNdEA, respectively) were increased when assayed with this mutant, with no significant differences in CYP2A6 bioactivation of NNK. Seemingly, the effect of CPR mutant I245P was CYP isoform and substrate dependent. The bioactivation capacity of all three CYPs was consistently augmented when assayed with CPR mutant R246A, i.e., all tested compounds demonstrated increased mutagenicity levels, in comparison with CYPs sustained by WT CPR. Mutant S243P demonstrated no significant differences in the bioactivation capacity of the three CYPs for the tested compounds, when compared with WT CPR.

#### 2.2.2. Membrane Preparations

##### CYP-Enzyme Kinetic Analysis

Enzyme activities of CYP1A2, 2A6 and 3A4 were measured using specific fluorogenic probe substrates ethoxyresorufin (EthR), coumarin and dibenzylfluorescein (DBF), respectively). Reaction velocities could be plotted according to the Michaelis-Menten equation and kinetic parameters (*k*_cat_ and *K*_M_) could be derived (Table 3). In general, CYP activities promoted by CPR mutants showed lower turn-over rates (*k*_cat_) and affinities (*K*_M_) when compared with WT that were also CYP form dependent. However, differences were minor in both constants and *k*_cat_/*K*_M_ values were not significantly different from WT values for all tested CYPs (Table 3).

##### Ionic Strength Effect on CYP:CPR Interaction

We previously demonstrated that mutations in the hinge region of human CPR strongly influences ionic strength profiles of ET to cytochrome *c* [24]. Ionic strength dependency of CPR’s ET was thus analyzed via CYP activities measurement in presence of increasing NaCl concentrations (0–1.25 M) (Figure 2 and Figure 3). Control experiments showed no effect of the salt concentration on the fluorescence of the formed products (Appendix A) or on the pH of the reaction mixture (data not shown). Interestingly, the maximum velocities of the three CYPs seem to occur at lower ionic strength conditions for all CPR (WT or variants) when compared to their maximum velocities in cytochrome *c* reduction reported in our previous study [24].

The salt dependence of CYP1A2 activity demonstrated a bell-shaped EthR O-deethylase activity curve with all CPR tested (Figure 2A), analogous to the ones described in our former study when measuring cytochrome *c* reduction [24]. The maximum activity (maximum *k_obs_*) was obtained, on average, at 100 mM NaCl. CYP1A2 activities dropped close to zero at the highest salt concentrations (1.25 M) for all CPR variants. CPR mutant S243P was more active than the WT, while other mutants showed lower activities when compared with WT.

For CYP2A6 (Figure 2B), the coumarin hydroxylase activity profiles showed less dependence on ionic strength as with CYP1A2 and showed no drop in activity at the highest salt concentrations. Maximum velocities were obtained approximately at 150 mM NaCl. All CPR mutants presented lower activities when compared with WT CPR.

CYP3A4 DBF O-debenzylase activity profiles (Figure 2C) were also bell-shaped, like CYP1A2, except for the CPR mutant R246A. The maximum activity was obtained, on average, at 100 mM NaCl. The activity dropped close to zero at the highest salt concentrations as was observed with CYP1A2. Interestingly, the CPR mutant R246A revealed a different salt profile with CYP3A4, leading to a continuous decrease of activity with increasing salt concentration, the maximal activity being even greater that the one obtained with WT CPR.

##### CYB5 Effect on Activity of CYP1A2, 2A6 and 3A4

CYB5 may act as an alternative donor of the second electron in the CYP catalytic cycle. We thus studied the effect of CPR hinge mutations when associated with CYB5, to determine if the presence of an alternative electron donor partner, capable of competing with CPR for CYPs is an important issue in the function of the hinge segment. The effects of CYB5 on CPR/CYP combinations were analyzed through enzyme activity assays in the presence of increasing concentrations of CYB5 and thus different CYB5:CPR ratios (Figure 4 and Table 4). Enzyme activities were normalized to the activity measured in the absence of CYB5. While the effect of CYB5 on CYP activities was not major, some interesting differences could be seen, notably in the CYB5 concentration giving the best stimulus. While for CYP1A2, the maximal effect was observed at 150 nM of CYB5, for CYP2A6 the concentration of CYB5 giving this maximal effect was dependent on the mutation, ranging from 50 nM with the WT CPR to 400 nM for the R246A mutant of CPR. For CYP3A4, the effect was relatively constant between all CPR mutants, but the concentration of CYB5 needed to obtain the maximal stimulation was much higher (400 nM). Overall, while CYP1A2 does not seem very sensitive to the presence of CYB5, a slight inhibition could be observed at high concentrations of CYB5. For the two other CYPs, the stimulation was quite pronounced, ranging from 2.6 to 3.4 or 1.8 to 2.6 for CYP2A6 and CYP3A4, respectively (Figure 4). No major differences were observed between CPR WT and mutants for each CPR/CYP combinations in term of the concentration of CYB5 to achieve the maximal effect, however, the intensity of the stimuli was different between CPR mutants.

## 3. Discussion

Microsomal CYP-mediated metabolism is dependent on ET through protein:protein interaction with its primary redox partner CPR. Traditionally, membrane anchoring and a negatively-charged surface patch of CPR have been considered to be major determinants of the proper alignment in this interaction, in which hydrophobic interactions have also been implicated [32,33]. Still, the view on CPR:CYP interaction for ET has become increasingly more complex with the recognition of CPR protein dynamics in ET, presenting an equilibrium between closed/locked and open/unlocked conformers in its function, in which the flexible hinge region seems to have a determinant role [10,16,17,18]. In a recent report we have shown the importance of this hinge segment and the effect of ionic strength in ET to the non-physiological redox partner cytochrome *c*, using eight different CPR mutants targeting four critical residues [24]. Data from this report lead us to hypothesize a hydrogen H-bond network around R246, important for the function of the open/closed dynamics.

Although the use of the soluble surrogate redox partner cytochrome *c* has been informative, the open/closed dynamics and thus ET may occur differently with CPR’s natural membrane-bound electron acceptors, such as CYPs. Three hinge mutants were selected from our former set, namely mutant R246A, for R246’s role in the suggested H-bond network, and mutants S243P and I245P demonstrating augmented cytochrome *c* reduction capabilities when compared with WT CPR. The CPR variants were combined with three different CYPs, representatives of three major CYP-families involved in drug metabolism. CYP families 1, 2 and 3 are responsible for 75% of all phase I metabolism of clinically used drugs: CYP3A4 is the major enzyme, and together these three CYPs are involved in almost 50% of metabolism of drugs [34]. Different experimental approaches were used to probe the effect of these hinge mutations.

The first general observation is that none of the three mutations caused a complete inactivation of CYP activity (i.e., obliterated CPR electron transfer to CYP), consistent with the data obtained using cytochrome *c* as electron acceptor [24]. We thus confirm that the three targeted positions in the hinge region are certainly part of a set of residues capable of controlling the CPR function and thus the effects of the mutations on internal and external ET are partially compensated by a larger network involved in the structural transitions (opening/closing).

The second interesting feature that these mutants demonstrated was their relative selectivity in inducing CYP-isoform dependent effects. This was first observed with CYP mediated bioactivation of pre-carcinogens, which seemed CYP isoform and in some cases substrate dependent. Still, the measurement of mutagenicity might be a quite indirect measurement of CYP activity, in generating DNA damaging metabolites. However, the specific features of CPR hinge mutants were more clearly demonstrated by the ionic strength dependency of CYP activities. Salt activity profiles were deviated differently by the three CPR hinge mutations, in a CYP-isoform dependent manner (Figure 2 and Figure 3). We previously demonstrated that the salt concentration at which the overall CPR to acceptor ET occurs depends on the relative difference in the rate of reduction of CPR and the rate of reduction of the electron acceptor [17]. The fact that all hinge mutations affect the salt concentration at which this maximal activity occur indicate that these CPR mutations modify the conformational equilibria. However, as noted above, all mutations have distinct salt-dependent signatures (Figure 3). We can thus hypothesize that the induced differences in conformational equilibria affect differently the three tested CYPs. Interestingly, all *V*_max_ and *K*_M_ values measured for the three CYPs were nearly identical between CPR mutants. This reinforces our current hypotheses addressing the conformational equilibria: The single hinge CPR mutants modify the salt-dependent conformational equilibria and not the intrinsic CYP properties. As such, they uncover the dependence of CYPs toward these conformational equilibria and provide a potential explanation of the mechanism by which CPR could, in certain conditions, favor one CYP over another.

An additional interesting feature of these salt profiles concerns the optimal salt concentration for CYP activity. Previously, when membrane-bound, WT CPR and the three mutants demonstrated very different salt concentrations for optimal cytochrome *c* reduction (ranging from 220–550 mM NaCl [24]). However, in the current study when combined with CYP1A2, 2A6 or 3A4, optimal activities were found at 100–200 mM NaCl, which together with the K/P reaction buffer approaches the ionic strength of physiological serum (154 mM KCl). This current data set exemplifies the difference in ET between the soluble surrogate electron acceptor cytochrome *c* and CPR’s natural membrane-bound redox partners. Maximum ET rates of soluble WT CPR in cytochrome *c* reduction was found to occur when CPR was equally distributed between open and closed conformers at approximately 375 mM NaCl [17]. With membrane bound WT CPR, maximum cytochrome *c* reduction rates were shifted above 500 mM NaCl, namely at 527 mM [24]. As such, data from our previous and current study suggests that optimal ET to various acceptors, either natural or artificial, occurs at very different ionic strengths. However, under physiological conditions, membrane bound CPR is present in an equilibrium, which is mostly in the locked state, however maintaining a very fast rate in alternating between open and closed conformations. This reinforce the idea that additional factors, such as the presence of membrane bound redox-partners, with stoichiometry’s favoring the electron acceptors (as is the case for CYPs, outnumbering CPR by a factor of 5–10), modulate the open/closed dynamics, as we put forward in our previous study [24].

As mentioned above, our data showed differences in the salt effect of the three different CYPs when combined with WT CPR, corroborating the study by Yun et al. [35] that ascribed the ionic strength effect observed mostly to CPR:CYP “interaction” while having relative minor effects on CYP protein conformation [36]. Still, results of the study of Voznesensky and Schenkman [37] indicated that charge pairing between CPR and CYP may not be the major determinant of the salt effect. Our current data,, as well as that of our previous study [24] confirms that the CPR:CYP interaction (i.e., electronic flow) dependency on ionic strength is mainly determined by its effect on the conformational equilibrium between locked and unlocked states of CPR and only in a minor manner by electrostatic interactions (affinity) between the FMN domain and the acceptor. Due to this major salt effect on CPR’s open/closed dynamics, the identification of potential electrostatic interactions between CPR and CYP have therefore been obscured. In retrospect, the seminal studies of Voznesensky and Schenkman [37,38] and of others (reviewed in [32]) on the salt effect of CYP catalysis in the quest for electrostatic interactions between CPR and CYP have been hampered by the lack of knowledge, at that time, on the salt-dependent protein dynamics of CPR.

In vivo, under constant ionic strengths conditions, affinity parameters may become determining in the CPR:CYP interaction, modulating the open/closed dynamics as indicated above. CYB5, the optional electron donor, demonstrated also stimulation/inhibition profiles that were dependent on CYPs, as well as on the various CPR used. This may also sign a relatively minor role for the affinity between CPR/CYP and CYB5/CYP interactions. Although CPR and CYB5 share similar, but not identical, binding-sites on the proximal side of CYP [39], the competition between the two electron donors does not seem to be a major factor in controlling CYP activities. Still, subtle differences in electrostatic interactions between the CPR/CYP and CYB5/CYP complexes have been shown [40,41], which seem even to be depending on substrate binding [42], indicating specific features in the sampling of the ensemble of open conformers of CPR.

Overall, our data point out that salt profiles are specific to CYP isoforms dependent interaction with CPR. The idea emerges for the existence of an ensemble of different unlocked CPR conformers that may be required for CYP-specific interactions. The highly flexible hinge region allows for a large ensemble of open conformations from which only a few or a subset may be required for ET to CYPs. CPR hinge mutants, by modifying the conformational equilibrium (as seen in salt profiles) may either promote or hinder specific conformations of the unlocked state, thus allowing or preventing interactions with (structurally) different redox partners. Such a selection of specific open conformations could explain the differential effect of the three hinge mutations on the activity of the different CYPs. Moreover, the soluble cytochrome *c* will sample these conformers quite differently when compared with the membrane-bound CYP redox partner, a plausible explanation for the difference in effects of the three hinge mutants when measuring ET to cytochrome *c* [24] or CYPs (this study).

From a structural perspective, it is clear that different CYP isoforms must share a common functional CPR binding surface. Although mitochondrial CYPs seem to have a signature of key basic amino acids on the proximal side for their interaction with the iron–sulfur protein adrenodoxin, such signature sequences do not exist for microsomal CYPs in their interaction with CPR [43]. This implies diversity in critical amino acids on their proximal side and suggests the possibility of affinity differences of microsomal CYPs for CPR, a plausible key element in the sampling of the open CPR conformers.

The conformational plasticity of CYPs could additionally play a role in this respect. Substrate binding has been shown to cause (subtle) conformational changes at the proximal site of CYPs, reviewed in Kandel and Lampe, 2014 [33], which may influence the affinity and thus sampling of open conformers of CPR for effective ET. In fact, this seems to be corroborated by our data of the hinge mutations, causing beside CYP-isoform dependent seemingly also substrate dependent effects, as described above.

It is tempting to speculate that CPR’s protein dynamics, containing different ensembles of closed and open conformations, was Nature’s way to enable CPR to be the “degenerated” electron supplier of so many (structural and functional) different redox-partners. In this respect it would be of interest to obtain insight on the evolutionary gain and thus the physiological relevance of such a universal electron donor for enzymes, involved in some many different and crucial metabolic pathways. In parallel to other central enzymes, the possibility may exist of a central hub- or central controller-function of CPR, for the fine tuning of multiple metabolic pathways and energy (NADPH) usage.

## 4. Materials and Methods

### 4.1. Reagents

l-Arginine, thiamine, chloramphenicol, ampicillin, kanamycin sulfate, isopropyl β-d-thiogalactoside (IPTG) (dioxane-free), δ-aminolevulinic acid, cytochrome *c* (horse heart), glucose 6-phosphate, glucose 6-phosphate dehydrogenase, nicotinamide adenine dinucleotide phosphate (NADP+ and NADPH), 2-aminoanthracene (2AA), *N*-nitrosodiethylamine (NNdEA), 4-(methylnitrosamino)-1-(3-pyridyl)-1-butanone (NNK), aflatoxin B1 (AfB1), resorufin, 7-hydroxy coumarine and fluorescein were obtained from Sigma-Aldrich (St. Louis, MO, USA). 2-amino-3-methylimidazo(4,5-f)quinoline (IQ) was obtained from Toronto Research Chemicals (North York, ON, Canada). LB Broth (Formedium, Norfolk, UK), bacto tryptone and bacto peptone were purchased from BD Biosciences (San Jose, CA, USA). Bacto yeast extract was obtained from Formedium (Norwich, England). EthR and coumarin were obtained from BD Biosciences (San Jose, CA, USA) and DBF from Santa Cruz Biotechnology (Santa Cruz, CA, USA). A polyclonal antibody from rabbit serum raised against recombinant human CPR obtained from Genetex (GTX101099) (Irvine, CA, USA) was used for immunodetection of the membrane-bound CPR.

### 4.2. Bacterial Expression of Human CYB5 and Purification

The cDNA of the open reading frame of human full length CYB5 was cloned in pET15b, as described in Nunez et al., 2010 [44], except that the full sequence was used instead of only the soluble part of it. The resulting plasmid was transformed into BL21-DE3 for expression. A single colony was grown in Terrific Browth medium containing 100 µg/mL ampicillin for 72 h with shaking at 22 °C. Cells were harvested by centrifugation at 4000× *g*, and the resulting pellet was resuspended and incubated for 30 min in 50 mM Tris-HCl, pH 7.4, containing 1 mM PMSF and 1 mg/mL lysozyme. Cells were lysed by sonication. Then 0.02 mg/mL RNase and 0.05 mg/mL DNase were added, and CYB5 was solubilized at 4 °C with 1% (*w*/*v*) sodium cholate, pH 7.4, for 1 h with moderate shaking. Supernatant was loaded onto a DEAE-cellulose anion-exchange column equilibrated with 0.2% (*w*/*v*) sodium cholate, 20 mM sodium/potassium phosphate buffer, pH 7.4. CYB5 was eluted with 0.5 M NaCl, 0.2% cholate, and 20 mM sodium/potassium phosphate buffer, pH 7.4. Fractions containing CYB5 were applied to a hydroxylapatite column equilibrated with 0.5 M NaCl and 20 mM sodium/potassium phosphate buffer, pH 7.4. Pure CYB5 was eluted with 0.1% (*w*/*v*) sodium cholate and 0.5 M sodium/potassium phosphate buffer and dialyzed against 0.1% (*w*/*v*) sodium cholate, 1 mM PMSF, and 20 mM sodium/potassium phosphate buffer. CYB5 content of samples was determined by spectrophotometric techniques as described previously [28,29]. CYB5 was concentrated and stored at −20 °C.

### 4.3. Bacterial Co-Expression of Human CPR Mutants and CYPs

CPR forms were expressed as a full-length membrane bound proteins using a dedicated *E. coli* host, the BTC strain, using the specialized bi-plasmid system adequate for co-expression of CPR with representative human CYP [26,27]. Plasmid pLCM_*POR* [20] was used for the expression of the membrane-bound, full-length WT form and mutants of human CPR [24]. The expression of human CYP-isoforms in the cell model was accomplished with plasmid pCWori containing human wildtype CYP cDNA (pCWh_1A2, pCWh_2A6 or pCWh_3A4) [45]. The pLCM_*POR* and the CYP plasmids pCWh were transfected through standard electroporation procedures [22]. Each strain was cultured in TB medium supplemented with peptone (2 g/L), thiamine (1 µg/mL), ampicillin (50 µg/mL), kanamycin (15 µg/mL), chloramphenicol (10 µg/mL), trace elements solution [46] (0.4 µL/mL), IPTG (0.2 mmol/L) and δ-Ala (100 µmol/L), final concentrations. Cultures were started with −80 °C glycerol stocks and cells were grown for 16 h at 28 °C with moderate agitation. CYP content of bacterial whole-cells preparations was determined using standard CO-difference spectrophotometry [47].

### 4.4. Whole-Cell Mutagenicity Assays

The mutagenicity assays were performed as described previously [20,23,45,47] using the liquid pre-incubation assay technique [48,49]). Briefly, BTC bacteria were grown for 18 h in TB medium supplemented with peptone (2 g/L), thiamine (1 µg/mL), ampicillin (50 µg/mL), kanamycin (15 µg/mL), a mixture of trace elements solution [46] (0.4 µL/mL) and IPTG (0.2 µmol/L), final concentrations. Pre-incubation was performed for 45 min in an orbital shaker at 37 °C before plating. Incubation buffer contained 10 mM glucose. Stock solutions of carcinogens were freshly made in dimethyl sulfoxide (DMSO) and working solutions were obtained by dilution in water. DMSO concentration in preincubations were ≤1.3%. Experiments were performed at least in triplicate. Revertant colonies on l-Arg selector plates were counted after 48 h incubation at 37 °C. Revertant colonies were determined by ProtoCOL 3 colony counter (Synbiosis, Cambridge, UK) using ProtoCOL V0 1.0.6 Software. CYP-mediated bioactivation was expressed in terms of mutagenic activity [l-arginine prototrophic (revertant) colonies per nmole of test compound, or in revertant colonies per μmole of test compound], determined from the slope of the linear portion of the dose–response curve.

### 4.5. Membrane Preparation and Characterization

Membrane preparations were isolated as previously described [22,24]. Briefly, cultures were harvested at 2772× *g* for 20 min at 4 °C. The pellet was resuspended in Tris-sucrose buffer (50 mM Tris-HCl, 250 mM sucrose, pH 7.8). Lysozyme was added to a final concentration of 0.5 mg/mL, EDTA (0.5 mM), phenylmethanesulfonyl fluoride (0.5 mM) and benzonase (12.5 U/mL). Cells were incubated on a roller bench for 30 min at 4 °C. Cell lyses was performed by freezing (−80 °C) and thawing (1 cycle) and by subsequent several short rounds (30 s) of low-intensity sonication, interspersed with 60 s of ice-bath submersion. The suspension was centrifuged at 2772× *g*, for 20 min at 4 °C to eliminate unbroken cells. Membranes were pelleted by ultracentrifugation of the supernatant at 100,000× g at 4 °C for 60 min. Membranes were resuspended in TGE buffer (75 mM Tris-HCl, 10% (*v*/*v*) glycerol, 25 mM EDTA, pH 7.5) using a Potter homogenizer and stored at −80 °C. Protein concentrations were determined using the method described by Bradford, following the manufacturer’s protocol from Bio-Rad (San Francisco, CA, USA), using bovine serum albumin as the standard. CYP contents of membrane preparations were determined using CO-difference spectrophotometry. Membrane proteins were separated by SDS–PAGE gel electrophoresis (10% polyacrylamide gel) and either stained with Coomassie blue or electro-transferred to PVDF membranes and further processed. CPR content of membrane fractions was quantified by immunodetection against a standard curve of purified human, full-length WT CPR, using polyclonal rabbit anti-CPR primary antibody and biotin-goat anti-rabbit antibody in combination with the fluorescent streptavidin conjugate (WesternDot 625 Western Blot Kit; Invitrogen, Eugene, OR, USA) (see Appendix A). Densitometry of CPR signals was performed using LabWorks 4.6 software (UVP, Cambridge, UK).

### 4.6. CYP-Enzyme Assays

Using membrane preparations, CYP-activities were assessed through determination of product formation by EthR O-deethylation (EROD; CYP1A2) (excitation 530 nm; emission 580 nm), coumarin 7-hydroxylation (CYP2A6) (excitation 330 nm; emission 460 nm) or *O*-debenzylation of DBF (CYP3A4) (excitation 485 nm; emission 535 nm) [20,22,26]. Assays were performed in 96-well format with a multi-mode microtiter plate reader (SpectraMax^®^i3x, Molecular Devices, USA) using SoftMax Pro 2.0 Software. Experiments were conducted with 8 nM CYP1A2, 100 nM CYP2A6, and 25 nM CYP3A4 (final well concentrations). Reactions were performed in 100 mM potassium phosphate buffer (without NaCl) (pH 7.6) supplemented with 3 mM MgCl_2_ and an NADPH regenerating system (NADPH 200 µM, glucose 6-phosphate 500 µM and glucose 6-phosphate dehydrogenase 40 U·L^−1^, final concentrations). Stock solutions of EthR were prepared in DMSO, while coumarin and DBF were prepared in acetonitrile (ACN). Final solvents concentrations were maintained constant throughout the experiment (0.2% (*v*/*v*) DMSO or 0.1% (*v*/*v*) ACN). Product formation was followed for 10 min at 37 °C and rates were calculated by using a standard curve of the products. Reactions were performed in triplicate with substrate concentrations ranging up to 5 µM EthR (CYP1A2), 20 µM coumarin (CYP2A6) or 10 µM DBF (CYP3A4). Velocity data were plotted according to the Michaelis–Menten equation with high confidence (*r*^2^ > 0.95) using GraphPad Prism 5.01 Software (La Jolla, CA, USA) and kinetic parameters (*k*_cat_ and *K*_M_) were derived [20,22,47]. Variance in data was analyzed using one-way ANOVA with Bonferroni’s multiple comparison tests, with 95% confidence interval—GraphPad Prism 5.01 Software (La Jolla, CA, USA).

### 4.7. Ionic Strength Effect

Catalytic activity of CYP1A2, 2A6, and 3A4, sustained by WT CPR and CPR mutants, was assessed at various NaCl concentrations (0–1.25 M), using 5 µM EthR, 20 µM coumarin and 10 µM DBF, respectively, in 100 mM potassium phosphate buffer (pH 7.6), and NADPH regenerating system (NADPH 200 µM, glucose 6-phosphate 500 µM and glucose 6-phosphate dehydrogenase 40 U L^−1^, final concentrations). Velocities were measured in triplicate in 96-well format using multi-mode microtiter plate reader (SpectraMax^®^i3x, Molecular Devices, San José, CA, USA; SoftMax Pro 2.0). Initial rates (picomoles of fluorescent product formed per picomoles of CYP per minute) were derived from the linear part of the kinetic traces using a standard curve of the respective products. Control experiments were conducted to assess the effect of ionic strength on the pH of the reaction matrix and the fluorescence of products.

### 4.8. CYP Activity Titration with CYB5

The CYB5 titration assay was performed in microplate format (96 wells) using the same enzyme assay conditions as described above (without NaCl), except substrate concentrations were hold constant (5 µM EthR, 20 µM coumarin or 7.5 µM DBF), applying a gradient of CYB5 (0–2000, 0–800 and 0–1000 nM for CYP1A2, 2A6 and 3A4, respectively).

## Figures and Tables

**Figure 1 ijms-19-03914-f001:**
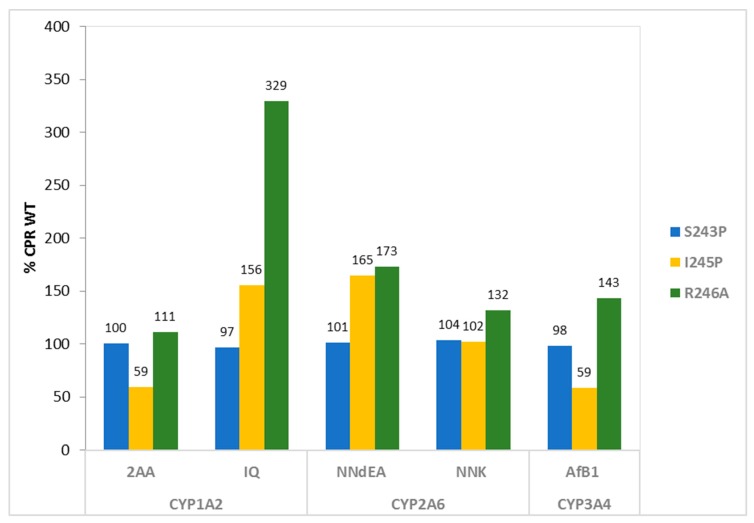
Bioactivation of pre-carcinogens mediated by CYP 1A2, 2A6 or 3A4 when combined with the three CPR hinge mutants. Bioactivation capacities were normalized with the one observed when CYPs were combined with WT CPR. (2AA: 2-aminoanthracene; IQ: 2-amino-3-methylimidazo(4,5-*f*)quinoline; NNdEA: *N*-nitrosodiethylamine; NNK: 4-(methylnitrosamino)-1-(3-pyridyl)-1-butanone; AfB1: aflatoxin B1).

**Figure 2 ijms-19-03914-f002:**
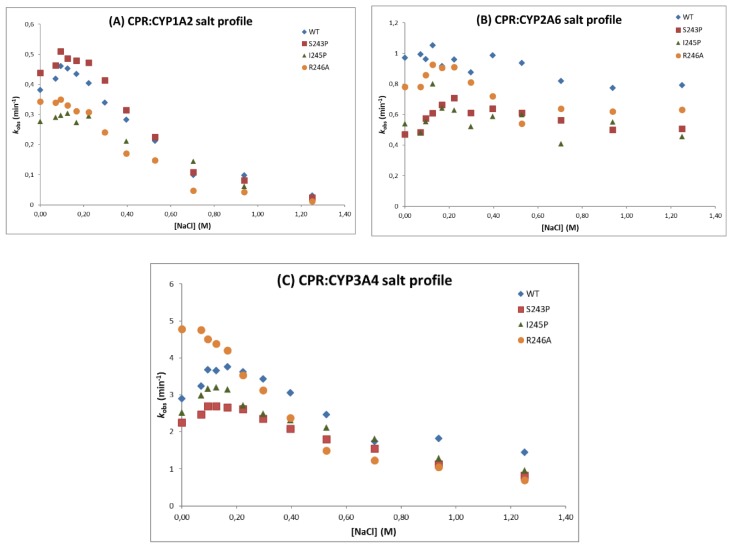
CYP reaction velocity (*k*_obs_) in function of the NaCl concentration, for the WT and mutant forms of CPR with CYP1A2 (**A**), CYP2A6 (**B**) or CYP3A4 (**C**).

**Figure 3 ijms-19-03914-f003:**
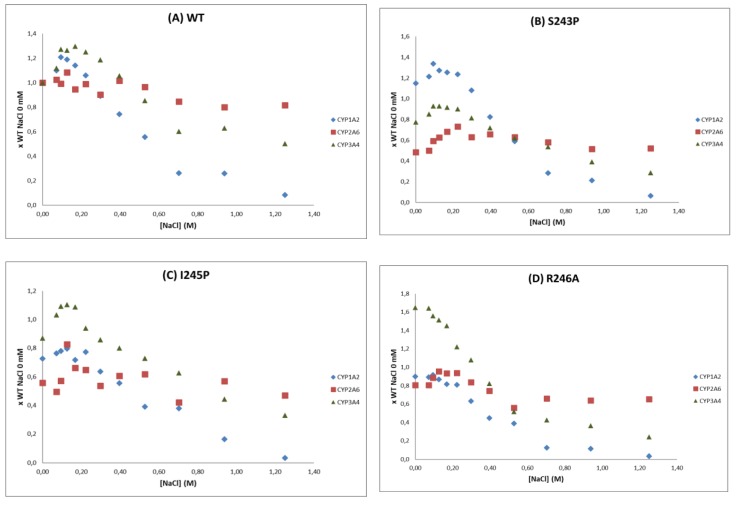
Relative CYP-reaction velocities (*k*_obs_) of CYP1A2, 2A6 and 3A4 in function of the NaCl concentration, when combined with CPR WT (**A**) or mutants S234P (**B**), I245P (**C**) and R246A (**D**). Velocities were normalized with the one observed when combined with WT CPR at 0 M NaCl.

**Figure 4 ijms-19-03914-f004:**
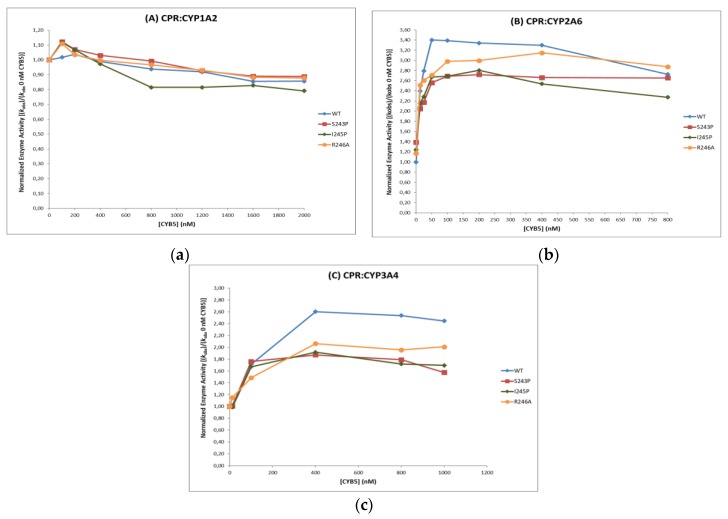
Effect of CYB5 concentration on maximum reaction velocities of CYP1A2 (**A**), 2A6 (**B**) and 3A4 (**C**) when sustained by WT CPR and the three hinge mutants. Activities were normalized to the one without CYB5 for each CPR mutant in comparison to WT.

**Table 1 ijms-19-03914-t001:** Microsomal cytochrome P450 (CYP) and NADPH cytochrome P450 oxidoreductase (CPR) contents of BTC cultures and membrane fractions.

CYP Isoform	CPR Form	Whole-Cells	Membrane Fractions		
		CYP ^1^	CYP ^1^	CPR ^1^	CPR:CYP Ratios
		(nM)	(pmol/mg Protein)		
**CYP1A2**	WT	109 ± 4	54 ± 1	4.1 ± 1.5	1:13
	S243P	241 ± 4	73 ± 4	7.7 ± 0.2	1:9
	I245P	206 ± 11	102 ± 1	6.1 ± 0.5	1:17
	R246A	176 ± 3	91 ± 2	5.4 ± 0.2	1:17
**CYP2A6**	WT	130 ± 2	139 ± 1	10.5 ± 1.3	1:13
	S243P	98 ± 1	106 ± 3	11.2 ± 0.5	1:9
	I245P	96 ± 7	102 ± 1	9.5 ± 0.9	1:11
	R246A	98 ± 2	146 ± 1	10.6 ± 1.5	1:14
**CYP3A4**	WT	105 ± 2	83 ± 3	19.8 ± 0.2	1:4
	S243P	122 ± 3	77 ± 2	22.5 ± 2.6	1:3
	I245P	128 ± 3	78 ± 2	18.3 ± 0.7	1:4
	R246A	143 ± 5	78 ± 1	21.4 ± 0.3	1:4

^1^ CYP and CPR contents are mean ± sd.

**Table 2 ijms-19-03914-t002:** CYP-mediated bioactivation of pre-carcinogens.

CYP Isoform	Mutagen	CPR Form			
		WT	S243P	I245P	R246A
**CYP1A2**	2AA ^1^	5643 ± 271	5666 ± 177	3339 ± 145	6274 ± 106
	IQ ^1^	335 ± 8	323 ± 7	521 ± 104	1103 ± 253
**CYP2A6**	NNdEA ^2^	537 ± 14	543 ± 21	885 ± 122	929 ± 118
	NNK ^2^	770 ± 121	799 ± 110	788 ± 57	1014 ± 29
**CYP3A4**	AfB1 ^1^	1129 ± 97	1109 ± 115	661 ± 185	1616 ± 163

Values are mean ± sd of three independent experiments, expressed as the number of revertant colonies per nmol ^1^ or per µmol ^2^ of pre-carcinogen.

**Table 3 ijms-19-03914-t003:** Michaelis-Menten kinetic parameters of CYP activities.

CYP Isoform	CPR Form	*k* _cat_ ^1^	*K* _M_ ^1^	Efficiency
		(Product Formed pmol·min^−1^·pmol^−1^ CYP)	(µM)	(*k*_cat_/*K*_M_) (% WT)
**CYP1A2**	WT	0.62 ± 0.02	1.94 ± 0.16	0.32 (1.00)
	S243P	0.63 ± 0.01	1.23 ± 0.05	0.51 (1.59)
	I245P	0.40 ± 0.01	0.89 ± 0.04	0.46 (1.44)
	R246A	0.43 ± 0.01	0.74 ± 0.06	0.59 (1.84)
**CYP2A6**	WT	1.37 ± 0.07	1.99 ± 0.34	0.69 (1.00)
	S243P	1.17 ± 0.08	2.03 ± 0.40	0.58 (0.84)
	I245P	1.20 ± 0.09	1.78 ± 0.39	0.67 (0.97)
	R246A	1.53 ± 0.08	1.98 ± 0.33	0.77 (1.12)
**CYP3A4**	WT	2.53 ± 0.16	3.75 ± 0.55	0.67 (1.00)
	S243P	1.83 ± 0.08	3.07 ± 0.34	0.60 (0.88)
	I245P	1.89 ± 0.13	2.85 ± 0.50	0.66 (0.98)
	R246A	2.18 ± 0.22	3.82 ± 0.87	0.57 (0.85)

^1^*k*_cat_ and *K*_M_ values are expressed as mean values of three independent experiments ± sd, determined at 0 M NaCl.

**Table 4 ijms-19-03914-t004:** Effect of CYB5 on CPR/CYP combinations.

CYP Isoform	CPR Form	Maximal *k_obs_* ^1^	CYB5 Stimulus ^2^	CYB5:CPR Ratio ^3^
			(%)	
**CYP1A2**	WT	1.04 ± 0.01	100	163
	S243P	1.11 ± 0.02	107	118
	I245P	1.12 ± 0.04	108	208
	R246A	1.12 ± 0.02	108	211
**CYP2A6**	WT	3.40 ± 0.04	100	26
	S243P	2.72 ± 0.05	80	106
	I245P	2.69 ± 0.04	79	43
	R246A	3.15 ± 0.04	93	211
**CYP3A4**	WT	2.60 ± 0.01	100	66
	S243P	1.87 ± 0.02	72	55
	I245P	1.92 ± 0.01	74	68
	R246A	2.06 ± 0.02	79	67

^1^ Normalized maximum enzyme activity: (*k*_obs_)/(*k*_obs_ at 0 nM CYB5). Values are expressed as the mean of three independent experiments ± sd. ^2^ Maximum CYB5 stimulus of enzyme activity when CYPs were combined with mutant CPRs, relative to the ones obtained with CPR WT. ^3^ The ratio at which maximum *k*_obs_ was reached. Grey shaded values: Maximum fold stimulation of enzyme activity by CYB5 when CYPs were combined with WT CPR.

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
