# Peer review of "Probing the Role of the Hinge Segment of Cytochrome P450 Oxidoreductase in the Interaction with Cytochrome P450"

_ijms, 2018, doi:10.3390/ijms19123914_

Round 1
Reviewer 1 Report
The paper of Campelo et al. presents a study on the interaction between Cytochrome P450 and its NADPH cytochrome P450 reductase.
The paper is well designed and of good scientific content.
However I have some critics on the form.
1) For instance I believe that figure 1 is difficult to understand. I would have prefered that the authors use a 0-400% scale and use the percent values, not percent value minus 100.
This would be clearer for general scientist (perhaps not specialists). Concerning the strain used for measuring the mutagenic power of the P450 mix, it would be nice to give a more complete description and references. People like me are familiar with the Ames system , the umuc system or the SOS system. Not the one used here. You must also define the initials used in the legend of the figure (first apparition in the paper) since their definition only appear later in the experimental at the end of the paper.
2) In table 4 : I dont understand the Cytochrome b5 (CYB5) stimulus. Is it the value over 100%
What do you mean? On figure 4 we have as high as 3.5 time stimulation by CYB5, but in the table the highest value is 108% thus probably 2.08 time? This must be clarified. You have plenty of space. Thus give longer legends with more explanations
3) page 8 paragraph line 243: Is it possible that the salt displaces partlially the ionic interaction between the CYP and the reductase. Also 100 mM phosphate ph 7.4 has allready an ionic strengh of about 0.2. Thus the addition of NaCl increases the ionic strengh but only starting from 0.2. Often when one study ionic strengh one use 50 mM HEPES buffer then add NaCl or KCl. Clarify.
I also agree with you that ionic strengh variations are a test tube study.
4) Concerning the discussion I wonder why you did not mention the studies by Lucie Waskell group and Ramamoorthy on b5_Reductase interactions (NMR and StopFlow)
As a whole the paper is interesting, and scientifically sound. The finding of differences between wild type reductase and some natural mutants is relevant to some human patients. The paper should be published after a number of small corrections n
Author Response
Response to Reviewer 1 Comments
Point 1:
1a. For instance I believe that figure 1 is difficult to understand. I would have prefered that the authors use a 0-400% scale and use the percent values, not percent value minus 100. This would be clearer for general scientist (perhaps not specialists).
Response 1a:
We have modified Figure 1 according to the indications of the reviewer (page 4);
1b. Concerning the strain used for measuring the mutagenic power of the P450 mix, it would be nice to give a more complete description and references. People like me are familiar with the Ames system , the umuc system or the SOS system. Not the one used here.
Response 1b:
We have modified the text of section 4.4, with inclusion of more details and references describing the mutagenicity assay with the BTC-CYP strains (pages 11-12;
1c. You must also define the initials used in the legend of the figure (first apparition in the paper) since their definition only appear later in the experimental at the end of the paper.
Response 1c:
We have now included the designation of the abbreviations in the legend of Figure 1 (page 5); as well as in the text of Section 2.2.1 (page 3, lines 104-106)
Point 2: In table 4 : I dont understand the Cytochrome b5 (CYB5) stimulus. Is it the value over 100%. What do you mean? On figure 4 we have as high as 3.5 time stimulation by CYB5, but in the table the highest value is 108% thus probably 2.08 time? This must be clarified. You have plenty of space. Thus give longer legends with more explanations
Response 2:
We have adapted Table 4 (grey shading) as well its legend to clarify and make the Table clearer (page 8, lines 197-200);
Point 3)
3a. page 8 paragraph line 243: Is it possible that the salt displaces partlially the ionic interaction between the CYP and the reductase.
Response 3a:
Actually, we are of the opinion that salts may partially displace the ionic interaction between CYP and CPR, as is indicated in the Discussion section, lines 273-275;
3b. Also 100 mM phosphate ph 7.4 has allready an ionic strengh of about 0.2. Thus the addition of NaCl increases the ionic strengh but only starting from 0.2. Often when one study ionic strengh one use 50 mM HEPES buffer then add NaCl or KCl. Clarify.
Response 3b:
The Reviewer was absolutely right in pointing out the additional ionic strength present in the reaction mixture by the used K/P buffer. We have adapted the text in the Discussion section now including this issue (page 9, lines 253-254)
Point 4: Concerning the discussion I wonder why you did not mention the studies by Lucie Waskell group and Ramamoorthy on b5_Reductase interactions (NMR and StopFlow)
Response 4:
We have now included references in the Discussion section, to mention the works of Dr Lucy Waskell and Dr. Ayyalusamy Ramamoorthy (page 10, lines 287-290)

Reviewer 2 Report
The authors of the manuscript studied three P450 reductase mutants S243P, I245P and R246A using CYP1A2, 2A6 and 3A4. These hinge mutations influenced the bio-activation of pre carcinogens based on the specific P450 used. Overall, the study enhances the understanding of the role of amino acids at the hinge region of CPR in the interaction with different CYPs.
The following are the two minor suggestions:
In the methods section, line 392, section 4.2; please change "Bacterial human CYB5 expression and purification" to Bacterial expression of human CYB5 and purification".
The authors should discuss in brief or in few sentences the method to express and purify the enzymes rather than citing previous papers. This include description in section 4.3, 4.3 and 4.5 of the methods section of the manuscript on pages 10 and 11.
Author Response
Response to Reviewer 2 Comments
Point 1: In the methods section, line 392, section 4.2; please change "Bacterial human CYB5 expression and purification" to Bacterial expression of human CYB5 and purification"
Response 1: The suggested alteration of the title of section 4.2 has been incorporated (page 11, line 338)
Point 2: The authors should discuss in brief or in few sentences the method to express and purify the enzymes rather than citing previous papers. This include description in section 4.3, 4.3 and 4.5 of the methods section of the manuscript on pages 10 and 11.
Response 2: More detailed descriptions of procedures of sections 4.2, 4.3 and 4.5 have been incorporated. page 11, lines 339-335, page 11, lines 357-368 and page 12, lines 384-404 respectively;
